# Protection of Chickens against H9N2 Avian Influenza Isolates with a Live Vector Vaccine Expressing Influenza Hemagglutinin Gene Derived from Y280 Avian Influenza Virus

**DOI:** 10.3390/ani14060872

**Published:** 2024-03-12

**Authors:** Jun-Feng Zhang, Sang-Won Kim, Ke Shang, Jong-Yeol Park, Yu-Ri Choi, Hyung-Kwan Jang, Bai Wei, Min Kang, Se-Yeoun Cha

**Affiliations:** 1Department of Avian Diseases, College of Veterinary Medicine and Center for Avian Disease, Jeonbuk National University, Iksan 54596, Republic of Korea; jfzhang018@gmail.com (J.-F.Z.); sk970221@gmail.com (S.-W.K.); shangke0624@gmail.com (K.S.); jyp0410@jbnu.ac.kr (J.-Y.P.); 9cinderella7@naver.com (Y.-R.C.); hkjang@jbnu.ac.kr (H.-K.J.); weibai116@hotmail.com (B.W.); 2College of Medical Technology and Engineering, Henan University of Science and Technology, Luoyang 471003, China; 3College of Animal Science and Technology, Luoyang Key Laboratory of Live Carrier Biomaterial and Animal Disease Prevention and Control, Henan University of Science and Technology, Luoyang 471000, China; 4Bio Disease Control (BIOD) Co., Ltd., Iksan 54596, Republic of Korea

**Keywords:** turkey herpesvirus, CRISPR/Cas9, NHEJ, H9N2, Y280, HA protein, vaccine efficacy

## Abstract

**Simple Summary:**

The H9N2 subtype of avian influenza virus (AIV) was first identified in turkey flocks in Wisconsin, USA, in 1966. Since then, H9N2 AIVs have been detected in mammals, domestic poultry, and wild birds. In June 2020, a new H9N2/Y280 was discovered in Korea. Existing commercial vaccines might not be fully effective against H9N2/Y280. Since influenza viruses undergo mutations rapidly, fast and accurate vaccine production is critical to address variant strains. Consequently, this study focused on rapid and effective vaccine production techniques against H9N2/Y280. The recombinant rHVT/Y280 was successfully constructed using NHEJ-CRISPR/Cas9 gene-editing technology. rHVT/Y280 vaccination induced strong humoral immunity, providing significant protection against H9N2/Y280 strain A21-MRA-003 following a challenge of 2 × 10^7^ 50% egg infectious dose. Therefore, rHVT/Y280 is a potential candidate for a live vaccine against H9N2/Y280.

**Abstract:**

Since the outbreak of the H9N2/Y439 avian influenza virus in 1996, the Korean poultry industry has incurred severe economic losses. A novel possibly zoonotic H9N2 virus from the Y280-like lineage (H9N2/Y280) has been prevalent in Korea since June 2020, posing a threat to the poultry sector. Rapid mutation of influenza viruses urges the development of effective vaccines against newly generated strains. Thus, we engineered a recombinant virus rHVT/Y280 to combat H9N2/Y280. We integrated the hemagglutinin (*HA*) gene of the H9N2/Y280 strain into the US2 region of the herpesvirus of turkeys (HVT) Fc126 vaccine strain, utilizing CRISPR/Cas9 gene-editing technology. The successful construction of rHVT/Y280 was confirmed by polymerase chain reaction and sequencing, followed by efficacy evaluation. Four-day-old specific pathogen-free chickens received the rHVT/Y280 vaccine and were challenged with the H9N2/Y280 strain A21-MRA-003 at 3 weeks post-vaccination. In 5 days, there were no gross lesions among the vaccinated chickens. The rHVT/Y280 vaccine induced strong humoral immunity and markedly reduced virus shedding, achieving 100% inhibition of virus recovery in the cecal tonsil and significantly lowering tissue viral load. Thus, HVT vector vaccines expressing HA can be used for protecting poultry against H9N2/Y280. The induction of humoral immunity by live vaccines is vital in such cases. In summary, the recombinant virus rHVT/Y280 is a promising vaccine candidate for the protection of chickens against the H9N2/Y280.

## 1. Introduction

Influenza viruses are categorized into three types based on antigenic differences in nucleocapsid (NP) and matrix (M) proteins: A, B, and C. The avian influenza virus (AIV) is an important subtype of influenza A virus [1]. Belonging to the Orthomyxoviridae family, AIV is an enveloped, negative-sense, single-stranded RNA virus. Its genome is segmented into eight distinct genes, encoding two significant surface glycoproteins, hemagglutinin (HA) and neuraminidase (NA), among others. The antigenic diversity of HA and NA is a foundation for AIV classification into 16 HA (H1–H16) and 9 NA (N1–N9) subtypes. Furthermore, AIV is categorized into highly pathogenic AIV (HPAIV) and low pathogenic AIV (LPAIV) based on pathogenicity in chickens. HPAIV primarily includes H5 and H7 strains, exemplified by H5N1 and H7N9, while LPAIV is mainly represented by H9N2 [2].

The H9N2 subtype of AIV was first identified in 1966 in turkey flocks in Wisconsin, USA [3]. Subsequently, H9N2 AIVs were detected in mammals, domestic poultry, and wild birds [4,5,6]. As the most prevalent influenza virus subtype in poultry, H9N2 AIV poses significant economic challenges and public health risks. H9N2 AIVs are generally classified into Eurasian and American lineages [7,8]. Within the Eurasian lineage, three sublineages are recognized: the G1-like lineage (A/quail/Hong Kong/G1/1997; H9N2/G1), the Y280-like lineage (A/duck/Hong Kong/Y280/1997; H9N2/Y280), and the Y439-like lineage (A/duck/Hong Kong/Y439/1997; H9N2/Y439) [9]. The American lineage of H9N2 AIVs is predominantly found in wild birds. Moreover, H9N2 AIVs can be transmitted from avian species to mammals and humans [10]. Additionally, other emerging AIVs, such as H5N2, H6N1, H7N7, H7N9, and H10N8, might have acquired gene segments from H9N2 AIVs [11,12,13], emphasizing the importance of controlling H9N2 virus spread to safeguard public health.

Korea experienced its first H9N2 AIV outbreak in 1996, with the H9N2/Y439 as the causative strain. This lineage subsequently became predominant, significantly impacting the Korean poultry industry [14]. In response, the Korean government has sanctioned the use of an inactivated H9N2 vaccine since 2007. Additionally, enhanced biosecurity measures and consistent surveillance in live bird markets (LBMs) were implemented to manage H9N2 prevalence. These interventions successfully mitigated the H9N2 impact on broilers and layers [15]. Nonetheless, the continued spread of H9N2 AIV among Korean native chickens (KNCs) in LBMs facilitated H9N2/Y439 reassortment with LPAIV from Eurasian aquatic birds, resulting in novel H9N2/Y439 with varied pathogenicity [16,17]. In June 2020, a new H9N2/Y280 was detected in KNCs during national surveillance [18]. Furthermore, existing commercial vaccines are not fully effective against H9N2/Y280 [19]. Therefore, the development of a live vaccine capable of protecting against H9N2/Y280, stimulating humoral immune responses, and adapting to evolving virus strains is imperative.

Herpesvirus of turkeys (HVT) has been extensively used as a vaccine for preventing Marek’s disease (MD) in chickens [20]. HVT is also used as a viral vector for expressing heterologous antigens of various avian diseases, such as avian influenza, Newcastle disease, infectious bursal disease, and infectious laryngotracheitis (ILT) [21,22]. Generally, recombinant HVT-vectored vaccines stimulate strong humoral immune responses and have high protective efficacy against target diseases [23,24,25]. Many methods, including traditional homologous recombination, overlapping cosmids, and bacterial artificial chromosome (BAC) clones, facilitated the construction of recombinant HVT [26]. However, these methods present several disadvantages, such as being labor-intensive and time-consuming.

The recent advent of the CRISPR (clustered regularly interspaced short palindromic repeats)/Cas9 system has brought a paradigm shift in genome editing, offering novel avenues for genetic modification [27,28,29]. The core components of the type II CRISPR/Cas system include an RNA-guided Cas9 endonuclease, initially discovered in various bacterial species (*Streptococcus pyogenes*), a single guide RNA (sgRNA), and a trans-activating crRNA (tracrRNA). The sgRNA directs the Cas9 protein to a specific 20-nucleotide target sequence next to a 5′-NGG-3′ protospacer adjacent motif (PAM) [30,31]. Cas9 binds to this sequence and generates a double-strand break (DSB). These DSBs are then repaired through either homology-directed repair (HDR), a high-fidelity pathway, or the more error-prone non-homologous end-joining (NHEJ) mechanism [27]. CRISPR technology has been employed to edit mammalian cell genomes, modify genes in animal models, and manipulate the genomes of various DNA viruses, including herpes simplex virus type I, adenovirus, pseudorabies virus, vaccinia virus, Epstein–Barr virus, guinea pig cytomegalovirus, HVT, duck enteritis virus, and ILTV [32,33,34,35,36,37,38].

The increasing prevalence of the new H9N2/Y280 in poultry farms is attributed to the absence of an effective antigen-matched vaccine. The urgent need for a vaccine against the H9N2 variant underpins this study. We employed NHEJ-CRISPR/Cas9 technology to engineer a recombinant rHVT/Y280 containing the *HA* gene from the H9N2/Y280 strain. This gene was inserted into the US2 region of the HVT Fc126 strain. Then, the study evaluated the protective efficacy of this recombinant virus against a challenge with H9N2/Y280 in four-day-old specific pathogen-free (SPF) chickens.

## 2. Materials and Methods

### 2.1. Viruses and Cell Culture

The HVT Fc126 strain (GeneBank accession number AF291866.1) was preserved in liquid nitrogen in our laboratory. We prepared primary chicken embryo fibroblasts (CEFs) from 10-day embryos and maintained them in a 5% M199 medium (10×) (Gibco, Grand Island, NY, USA) supplemented with 5% fetal bovine serum (FBS) (Peak Serum, Wellington, CO, USA), 50% of F-10 (1×) nutrient mixture (Sigma, St. Louis, MO, USA), 1% antibiotic–antimycotic solution (Gibco, Grand Island, NY, USA), 10% tryptose phosphate broth (TPB) (Becton, Dickinson and Company, Sparks, MD, USA), and 7.5% sodium bicarbonate solution. The cultures were incubated at 37 °C in a 5% CO_2_ atmosphere.

The influenza virus H9N2/Y280 was isolated from organ samples (trachea, lung, and bronchus) collected in 2021 from a farm in Jeollabuk-do, Iksan City. The sampled birds exhibited respiratory symptoms, including cough, nasal discharge, and bronchitis. We performed a diagnostic polymerase chain reaction (PCR) using a primer set (P3_For: 5′-CATCCCAGTGCTGGGAAR GAYCCTAAGAA-3′; P3_Rev: 5′-AGAGCTCTTGTTCTCTGATAGGTG-3′). The protocol involved one cycle of pre-denaturation at 94 °C for 2 min, followed by 35 cycles of denaturation at 94 °C for 30 s, annealing at 58 °C for 30 s, and extension at 72 °C for 1 min, with a final extension step at 72 °C for 5 min. The strain identified from the isolated sample was designated as A21-MRA-003. Figure 1 shows the phylogenetic information of the isolated A21-MRA-003 strain.

### 2.2. Construction of Plasmids Expressing CAS9/gRNA and Donor Plasmids

The guide RNA (gRNA) targeting the US2 region of the HVT genome was designed using CHOPCHOP, a CRISPR gRNA design online tool (http://chopchop.cbu.uib.no/, accessed on 18 May 2022). The DNA oligo of gRNAs was synthesized and cloned into the CRISPR expression plasmid pSpCas9(BB)-2A-Puro (PX459) V2.0 (Addgene, Watertown, MA, USA), resulting in PX459-US2-gRNA. This cloning was achieved by inserting synthesized primers US2-gRNA-F/R into BbsI restriction sites [39]. In a similar process, the bait sequence sgB, which shares no homology with the genomes of humans, chickens, pigs, prokaryotic DNA sequences, or viruses, was synthesized and cloned into PX459, producing PX459-sgB-gRNA [40].

For the donor plasmid pGEM-sgB-LoxN-PacI-RFP-PacI-LoxN-SfiI-HA-SfiI-sgB, including red fluorescent protein (RFP) and H9N2/Y280-HA expression cassettes, sgB-LoxN-PacI-LoxN-SfiI-spacer-SfiI-sgB was synthesized by a commercial gene-synthesis service (Cosmogenetech, Seoul, Republic of Korea). This synthesized sequence was then cloned into the pGEM-T-easy vector, creating pGEM-sgB-LoxN-PacI-LoxN-SfiI-spacer-SfiI-sgB. Subsequently, the RFP expression cassette from mRFP1-C1 (Addgene, Watertown, MA, USA) was inserted into this vector through the PacI site, resulting in pGEM-sgB-LoxN-RFP-LoxN-SfiI-spacer-SfiI-sgB. Finally, the H9N2/Y280-HA-expressing cassette was incorporated into this construct using SfiI sites, forming the complete donor plasmid pGEM-sgB-LoxN-PacI-RFP-PacI-LoxN-SfiI-HA-SfiI-sgB. Primer sequences are detailed in Appendix A.

### 2.3. Generation of rHVT/Y280

The recombinant HVT (rHVT)/Y280 was generated using NHEJ-CRISPR/Cas9 gene-editing technology (Appendix A). Primary CEFs were plated in 24-well plates 48 h before transfection at a density of 4 × 10^5^ cells/well. A mixture containing 0.25 μg PX459-US2-gRNA, 0.25 μg PX459-sgB-gRNA, and 0.5 μg of the donor plasmid pGEM-sgB-LoxN-PacI-RFP-PacI-LoxN-SfiI-HA-SfiI-sgB was transfected into CEFs utilizing Lipofectamine 3000^®^ (Invitrogen, Carlsbad, CA, USA) following the manufacturer’s instructions. At 24 h post-transfection, the cells underwent puromycin selection for three days, followed by infection with HVT at a multiplicity of infection (MOI) of 0.01 plaque-forming unit (PFU)/cell. On the third day post-infection, half of the cells were subjected to PCR analysis to confirm gene editing, while the remainder were transferred to fresh CEFs for fluorescent plaque purification via two methods.

In the plaque digestion method using the RFP fluorescent markers in rHVT/Y280, a fluorescence filter set for Cyanine 3 (Cy3) in a fluorescent microscope was applied. Briefly, plaques in the phase contrast that overlaid with RFP fluorescent plaques by Cy3 filter sets were marked and subjected to digestion, followed by re-cultivation on fresh CEFs.

In the fluorescence-activated cell sorters (FACS) method, rHVT/Y280 was harvested and sorted into a 96-well plate pre-seeded with CEFs using the BD FACS Aria III cell sorter (Becton, Dickinson and Company, Franklin Lakes, NJ, USA) with Cy3 excitation. RFP-positive rHVT/Y280-infected cells in each method were subsequently purified using three rounds of purification, and rHVT/Y280 purity was checked by PCR in every round of purification. Primer sequences are detailed in Appendix A.

### 2.4. Growth Kinetics

One hundred PFUs of rHVT/Y280 and parental HVT were simultaneously inoculated onto CEFs in 24-well plates to investigate the growth properties of rHVT/Y280. At 12, 24, 48, 72, 96, and 120 h post-infection, virus-infected CEFs were harvested. Viral DNA was extracted using the Viral RNA/DNA extraction kit (iNtRON) (intronbio, Seongnam-Si, South Korea) according to the manufacturer’s protocol. HVT genome copies per 10,000 cells were quantified using the Brilliant III Ultra-Fast SYBR Green QPCR Master Mix (Agilent Technologies; Santa Clara, CA, USA) according to the manufacturer’s protocol. SYBR Green-based real-time quantitative PCR (qPCR), taking the mean value for triplicate wells for each test sample, was performed using a primer sets (SORF1-F 5′-GGCAGACACCGCGTTGTAT-3′, SORF1-R 5′-TGTCCACGCTCGAGACTATCC-3′) for detection of the SORF1 region of HVT and a primer sets (ChOVO-F 5′-CACTGCCACTGGGCTCTGT-3′, ChOVO-R 5′-GCAATGGCAATAAACCTCCAA-3′) for detection of host ovotransferrin gene. The method used for qPCR was previously described [41,42].

### 2.5. Genetic Stability

The stability of rHVT/Y280 was assessed over 15 passages in CEFs. The presence of the inserted gene (*RFP-H9/Y280*) was verified at every fifth passage through PCR, using DNA samples extracted from the cells.

### 2.6. Preparation of Formalin-Inactivated Viruses

Formalin-inactivated vaccines were prepared from A21-MRA-003 CE2 (H9N2/Y280) and commercial vaccine strain A/chicken/Korea/01310/2001 (H9N2/Y439). Briefly, non-concentrated allantoic fluids of H9N2/Y280 and H9N2/Y439 were adjusted to 10^9.0^ 50% egg infectious dose (EID_50_)/0.1 mL. The viruses were inactivated by adding 0.1% formalin (*v*/*v*) and kept at 20 °C for 24 h [43]. The inactivation was confirmed by two blind passages in SPF eggs. The inactivated viruses were emulsified with oil (Montanide ISA70; SEPPIC, La Garenne-Colombes, France) at a ratio of 3:7 (*w*/*w*), and the final concentration (10^8.5^ EID_50_/0.1 mL) of inactivated H9N2/Y280 and H9N2/Y439 was used for immunization.

### 2.7. Chickens and Vaccination Programs

A total of 44 one-day-old SPF chickens were assigned to five groups to evaluate the protective efficacy of rHVT/Y280: A–C, positive control (PC), and negative control (NC) groups. Group A (*n* = 12) received subcutaneous immunization in the neck with 6000 PFUs of rHVT/Y280 in a 200 μL volume. Group B (*n* = 8) and group C (*n* = 8) received intramuscular vaccinations with inactivated H9N2/Y280 (10^8.5^ EID_50_/0.1 mL) and inactivated H9N2/Y439 (10^8.5^ EID_50_/0.1 mL), respectively. The PC group (*n* = 10) and the NC group (*n* = 6) received subcutaneous injections of PBS in a 200 μL volume in the neck. Chickens were bled weekly for 3 weeks post-vaccination (wpv) for serology testing. At 3 wpv, groups A, B, C, and PC were challenged intranasally with 400 μL of 2 × 10^7^ EID_50_ of A21-MRA-003 CE2 (H9N2/Y280). Chickens were observed daily for clinical signs and mortality. Oropharyngeal (OP) and cloacal (CL) swab samples were collected 3 and 5 days post-infection (dpi) to evaluate virus shedding. At the end of the experiment (5 dpi), all survived chickens were euthanized, necropsied, and examined for the presence of gross lesions. Furthermore, the lung, trachea, cecal tonsil (CT), brain, and bursa were collected for virus detection. Vaccine protection indices were calculated using the following formula:$$\begin{array}{l} inhibition\;of\;virus\;recovery\;rate\;against\;number\;of\;positive\;detections\;in\;CT\\ =100\%\text{–}positive\;detection\;rate\;of\;the\;vaccinated\;group/positive\;detection\;rate\;of\;the\;sham\;group. \end{array}$$

All experimental and animal management procedures were undertaken in accordance with the requirements of the Animal Care and Ethics Committee of Jeonbuk National University. The animal facility at Jeonbuk National University is fully accredited by the National Association of Laboratory Animal Care (approval number: NON2023-008).

### 2.8. Serology

Blood samples were obtained from each group on days 7, 14, and 21 after vaccinations. These samples were incubated at 37 °C for 1 h and then centrifuged at 3000 rpm for 15 min to separate the sera. Serum antibody titers against HA were quantified using the haemagglutinin inhibition (HI) assay according to standard protocol [44]. Briefly, two-fold dilutions of chicken serum samples were tested in duplicate in 96-well V-bottomed plates, followed by adding 4 hemagglutination units (HAU) of antigens (H9N2/Y280 and H9N2/Y439) that genetically and antigenically surrogate for the used vaccine strains and diluting in PBS. After the plates were incubated at room temperature for 0.5 h, 0.5% chicken red blood cells were added to the virus/serum mixture and incubated at room temperature for another 30 min. The HI antibody titer was determined as the reciprocal of the highest dilution that completely prevented red blood cells from agglutination.

### 2.9. Clinical Signs, Mortality, and Gross Lesions

The birds were checked daily for mortality and representative clinical signs (respiratory signs, depression, and diarrhea) in H9N2/Y280-infected chickens within 5 dpi. At 5 dpi, all surviving birds were subjected to autopsy, and the gross lesions of the internal organs (trachea, thymus, kidney, and bursa) were examined [9].

### 2.10. Assessment of Virus Shedding

OP and CL swab samples were collected on 3 and 5 dpi to evaluate virus shedding. Each swab sample was inoculated in the allantoic cavity of 10-day-old SPF embryonated eggs. Viral EID_50_ titers were determined by injecting 100 μL of serial 10-fold dilutions of the virus into the allantoic cavities of 10-day-old eggs. For each dilution, 4 eggs were utilized to ensure accurate titer estimation. The 50% endpoints were calculated using Reed and Muench’s method [26] for EID_50_ and are presented in log^10^ EID_50_/_mL_.

### 2.11. Virus Replication in Tissues

Tissue samples (lung, trachea, CT, brain, and bursa) were collected for viral detection at 5 dpi. The tissue samples were homogenized in 10% (*w*/*v*) PBS (pH 7.4; supplemented with 100× antibiotic–antimycotic solution (Gibco, New York, NY, USA)). The homogenates were centrifuged at 3000× *g* for 10 min at 4 °C, and the supernatant was then conserved in aliquots at −70 °C for virus titration.

### 2.12. Statistical Analysis

Statistical analysis was performed using SPSS version 21.0 (SPSS Inc., Chicago, IL, USA). Data were analyzed using a one-way analysis of variance (ANOVA). Statistical significance was defined at * *p* < 0.05, ** *p* < 0.01, and *** *p* < 0.001.

## 3. Results

### 3.1. Identification of PX459-US2-gRNA and PX459-sgB-gRNA

PCR was conducted using the PX459-US2-gRNA plasmid as a template. Primers U6 forward and US2-gRNA-R were employed, successfully amplifying a target band of 98 bp (Figure 2A). Sequencing confirmed the accurate construction of PX459-US2-gRNA (Figure 2C). Similarly, PCR was performed with primers U6 forward and sgB-gRNA-R using the PX459-sgB-gRNA template, resulting in the amplification of the same-sized target band (Figure 2B). Sequencing confirmed the successful construction of PX459-sgB-gRNA (Figure 2D).

### 3.2. Generation of rHVT/Y280

rHVT/Y280 was generated by co-transfecting CEFs with PX459-US2-gRNA, PX459-sgB-gRNA, and the donor plasmid pGEM-sgB-LoxN-PacI-RFP-PacI-LoxN-SfiI-HA-SfiI-sgB. At 24 h post-transfection, the cells were treated with puromycin for three days, followed by infection with HVT at an MOI of 0.01. Fluorescent plaque-expressing RFP, indicative of successful RFP-H9/Y280 insertion into the HVT genome, was observed at 3 dpi under the fluorescent microscope using the Cy3 filter sets (Figure 3). Four primer pairs were utilized for PCR identification. The primer pairs US2-F/RFP-internal-R and HA-internal-F/US2-R identified viruses inserted in the forward orientation, yielding target bands of 769 bp and 1042 bp, respectively (Appendix A). For viruses inserted in the reverse orientation, primer pairs US2-F/HA-internal-F and RFP-internal-R/US2-R were used with target bands of 449 bp and 1362 bp, respectively (Appendix A). Gel electrophoresis confirmed the expected results (Appendix A), and sequencing corroborated these findings (Appendix A). In summary, the RFP-H9/Y280 sequence was successfully integrated into the HVT genome in both forward and reverse orientations.

### 3.3. Purification of rHVT/Y280

Following three rounds of purification, rHVT/Y280 was identified using PCR. The primers US2-F/RFP-internal-R were employed for detecting the virus in the forward orientation, yielding a target band of 769 bp. Primers RFP-internal-R/US2-R were used to identify the virus in the reverse orientation, resulting in a target band of 1362 bp. For the detection of HVT presence, primers US2-F/US2-R were utilized, producing a target band of 869 bp. RFP-H9/Y280 was inserted in the forward direction in the final purified recombinant virus (Appendix A).

### 3.4. Growth Kinetics

The growth kinetics of rHVT/Y280 were measured to determine whether the RFP-H9/Y280 insertion into the US2 locus of parental HVT influenced the replication ability of the recombinant virus in vitro. The growth property between parental HVT and rHVT/Y280 was not significantly different (*p* > 0.05) through all tested time points. Thus, the rHVT/Y280 had a similar growth property to its parental HVT in CEFs (Figure 4).

### 3.5. Genetic Stability

The genetic stability of the *RFP-H9/Y280* gene was evaluated by serially passaging rHVT/Y280 in CEFs across 15 passages. PCR, conducted on viral DNA extracted every five passages, consistently showed amplification of the target band from the 5th, 10th, and 15th passage DNA samples of rHVT/Y280, indicating stable integration of the *RFP-H9/Y280* gene into the US2 locus of the HVT genome after 15 passages.

### 3.6. Humoral Immune Response

The immunogenicity of rHVT/Y280 in chickens was assessed by weekly collection and HI testing of sera post-vaccination. Two weeks post-vaccination (wpv), seropositivity rates for groups A, B, and C were 83%, 87%, and 100%, with corresponding antibody titers of 3.75 ± 2.01, 6.13 ± 2.75, and 3.75 ± 0.71, respectively. At 3 wpv, the seropositivity rates for all groups reached 100%, with mean antibody titers of 6.75 ± 1.22, 8.50 ± 1.51, and 6.63 ± 0.70, respectively. Therefore, rHVT/Y280 could induce a high level of humoral immunity (Figure 5).

### 3.7. Protective Efficacy in Chickens

Throughout the 5 dpi observation period, no clinical signs or mortality were observed in any group (Table 1). Necropsies were performed on all chickens at 5 dpi. The most severe gross lesions were noted in the PC group, with findings including hemorrhagic trachea (6/10), hemorrhagic thymus (6/10), swollen kidney (6/10), and hemorrhagic bursa (3/10). In group C (inactivated H9N2/Y439), the observed gross lesions were hemorrhagic trachea (4/8), hemorrhagic thymus (4/8), and swollen kidney (3/8). Groups A (rHVT/Y280), B (inactivated H9N2/Y280), and NC displayed no gross lesions (Appendix A). Virus shedding rates at 3 dpi were the highest in the PC group, with 100% (10/10) in the OP swabs and 80% (8/10) in the CL swabs. In group C, shedding rates were 100% (8/8) in OP and 75% (6/8) in CL swabs. Group A had lower shedding rates of 25% (3/12) in OP and 0% (0/12) in CL, followed by group B with a rate of 12.5% (1/8) in both OP and CL swabs. At 5 dpi, the PC group maintained high virus shedding rates of 100% (10/10) in OP and 90% (9/10) in CL swabs. Group C showed OP and CL shedding rates of 100% (8/8) and 37.5% (3/8), respectively. No H9N2/Y280 virus was detected in OP and CL swabs of groups A and B. Samples of trachea, lung, brain, bursa, and CT were collected at 5 dpi to assess virus replication in tissues. For the PC group, virus detection rates in the trachea, lung, brain, bursa, and CT were 100% (10/10), 70% (7/10), 60% (6/10), 50% (5/10), and 60% (6/10), respectively. The detection rates in group C were 87.5% (7/8) for the trachea, 62.5% (5/8) for the lung, 62.5% (5/8) for the brain, 12.5% (1/8) for the bursa, and 25% (2/8) for CT. Virus detection rates were considerably lower in groups A and B. In group A, the virus was detected only in the trachea and lung at a rate of 8.3% (1/12). In group B, detection in the trachea, lung, and brain was 12.5% (1/8). No virus was detected in all OP and CL samples or tissue samples in the NC group (Table 1). Based on the virus detection rate in the CT, the calculated protection indices for groups A, B, and C were 100%, 100%, and 58.3%, respectively. In summary, rHVT/Y280 (group A) and inactivated H9N2/Y280 (group B) conferred 100% protection against H9N2/Y280 infection, whereas inactivated H9N2/Y439 (group C) provided only partial protection.

## 4. Discussion

Globally, the H9N2 virus has become endemic across various countries and regions since its emergence in the 1990s. Characterized by its low pathogenicity, the H9 subtype of AIV typically induces only respiratory symptoms and a temporary reduction in egg production in chickens when it occurs as a single infection. However, its combination with other pathogens results in significantly increased morbidity and mortality [5,8]. This scenario not only gravely impacts the poultry industry but also presents considerable public health safety risks and a heightened threat to human health [4,6,10].

In South Korea, the Y439-like lineage has predominated since the initial H9N2 outbreak in 1996, with the virus being effectively managed through a commercial inactivated vaccine (H9N2/Y439, 01310). Nevertheless, a novel Y280-like lineage was identified in 2020 [8,18]. Existing commercial vaccines offer inadequate protection against this new lineage of H9N2, thereby introducing a fresh challenge to the poultry sector [19]. Consequently, there is a pressing need for the development of a vaccine specifically tailored to H9N2/Y280 to combat the H9N2/Y280 effectively. Therefore, this study utilized the *HA* gene from the newly identified Y280-like lineage to construct a recombinant virus.

Developing recombinant live vector vaccines that express key protective antigens of AIV has emerged as a significant research focus. HVT is a promising viral vector due to its extensive replication of non-essential regions, allowing the insertion of foreign genes [45,46]. As a live vector, HVT continuously infects chickens, leading to prolonged expression of foreign genes and sustained antibody production. It also elicits a robust and enduring cellular immune response [24,25,47]. A single inoculation with HVT can confer lifelong immunity [48]. It is non-pathogenic to chickens, does not impact their production, and is safe. HVT’s strong cell-binding capabilities facilitate cell-to-cell spread, enabling it to overcome maternal antibody interference [49,50,51]. It is also suitable for early immunization strategies, such as immunization of 1-day-old chickens or 18-day-old in ovo injections, to induce protective antibodies early [52]. Currently, several recombinant virus strains using HVT as a vector have been developed. Notably, the recombinant HVT live vaccine expressing the *IBDV-VP2* gene is now used in commercial production, offering effective immune protection for chickens [53].

CRISPR/Cas9 represents a significant advancement in gene-editing technology. Compared to earlier methods, its key attributes are simplicity, speed, and efficiency. This study focused on leveraging the CRISPR/Cas9 system for the swift development of a vaccine targeting the novel H9N2/Y280 strains. Previously, we utilized CRISPR/Cas9 to incorporate the *VP2* gene of IBDV into the UL45/46 region of the HVT genome, creating the recombinant virus rHVT-VP2 [54]. We meticulously fine-tuned the cellular conditions, concentrations of sgRNA and donor plasmids, and the transfection process. Under these optimized conditions, targeted red fluorescent plaques could be identified in a single trial. Therefore, our findings confirmed the high efficacy of CRISPR/Cas9 in editing the HVT genome. A primary objective was to develop a rapid platform for HVT vector vaccines using CRISPR/Cas9 to counter emerging poultry viruses. Furthermore, the ability of HVT to express only specific antigens facilitates the differentiation between vaccinated and field strains, offering a substantial advantage in practical vaccine deployment.

In this study, we constructed the rHVT/Y280 by integrating the *H9N2/Y280-HA* gene into the US2 region of the HVT Fc126 vaccine strain using CRISPR/Cas9-mediated gene editing. The insertion site critically influences foreign gene expression, as well as the growth and replication of the recombinant virus [46]. Over 20 non-essential regions for replication in HVT and MD virus (MDV) have been identified, with studied insertion sites including US2, US10, UL3-4, UL22-23, UL45-46, and US10-SORF [45,55]. Of these, US2 serves as an effective site for expressing vaccine targets, with superior performance compared to US10 [45]. Moreover, the growth kinetics of the recombinant virus rHVT-US2-HA were comparable to those of the parental HVT. Chickens immunized with rHVT-US2-HA exhibited enhanced protection and reduced mortality compared to those vaccinated with rHVT-US10-HA. Furthermore, rHVT-US2-HA immunization resulted in higher HI antibody titers than rHVT-US10-HA immunization. Overall, HVT expressing the *HA* gene at the US2 region provided significantly more effective protective immunity compared to the US10 region. Additionally, researchers inserted the H9N2 HA expression cassette into the US2, US10, and meq regions of the MDV genome. The expression levels of H9N2-HA at mRNA and protein levels were analyzed and compared using quantitative PCR and enzyme-linked immunosorbent assay (ELISA). Findings revealed that the HA expression cassette in the US2 region exhibited the highest transcriptional activity and protein expression [46]. Therefore, the US2 region was chosen as the insertion site for the *HA* gene in the current study.

Following the identification of the desired fluorescent plaques, the next phase involved their purification. This was achieved through plaque digestion and FACS methods. The plaque digestion method is non-expensive, digesting only fluorescent plaques. However, this method is time-consuming and can lead to the loss of the plaques during digestion. For these reasons, we applied the FACS method for the purification of rHVT/Y280. When compared with the plaque digestion method, the FACS method is simpler and more user-friendly. Previous studies applied these methods for the purification of rHVT and successfully obtained the purified rHVT from a mixing pool of rHVT and parent HVT [40,56]. Therefore, the FACS method is recommended to quickly obtain a purified recombinant virus.

Following the purification and proliferation of rHVT/Y280, we assessed its vaccine efficacy. In 3 wpv, the HI titer for rHVT/Y280 (6.8 ± 1.2) was comparable to that of the inactivated H9N2/Y439 (6.6 ± 0.7), but without significant difference (*p* > 0.05), and slightly lower than that of the inactivated H9N2/Y280 (8.5 ± 1.5). These findings suggest that rHVT/Y280 could induce robust humoral immunity.

Our findings indicate that virus shedding was absent in OP and CL samples from chickens vaccinated with rHVT/Y280 and inactivated H9N2/Y280 at 5 dpi. This suggests complete suppression of the virus in the upper respiratory and intestinal tracts. Conversely, the virus was detectable in most OP and CL samples in chickens immunized with inactivated H9N2/Y439, indicating that this inactivated vaccine was ineffective in preventing and controlling the virus. A similar pattern was observed in tissue samples, with the virus being undetectable in most samples from the rHVT/Y280 and inactivated H9N2/Y280 groups. However, in the inactivated H9N2/Y439 group, the virus was present in a significant portion of the tissue samples. These results demonstrate that inactivated H9N2/Y439 is no longer effective in providing protection against H9N2/Y280, primarily due to antigenic mismatch. Y280 represents a new lineage of H9N2, underscoring the necessity of developing new vaccines that match emerging viruses. In this study, the inactivated H9N2/Y280 vaccine provided complete protection, as did the rHVT/Y280 containing H9N2/Y280-HA.

## 5. Conclusions

In conclusion, the recombinant rHVT/Y280 was successfully constructed using the NHEJ-CRISPR/Cas9 gene-editing technology. Following vaccination, the rHVT/Y280 elicited strong humoral immunity, offering substantial protection against H9N2/Y280 strain A21-MRA-003 following a challenge of 2 × 10^7^ EID_50_. Thus, rHVT/Y280 effectively protected chickens from H9N2/Y280 infection, suggesting its potential as a live vaccine candidate against H9N2/Y280.

## Figures and Tables

**Figure 1 animals-14-00872-f001:**
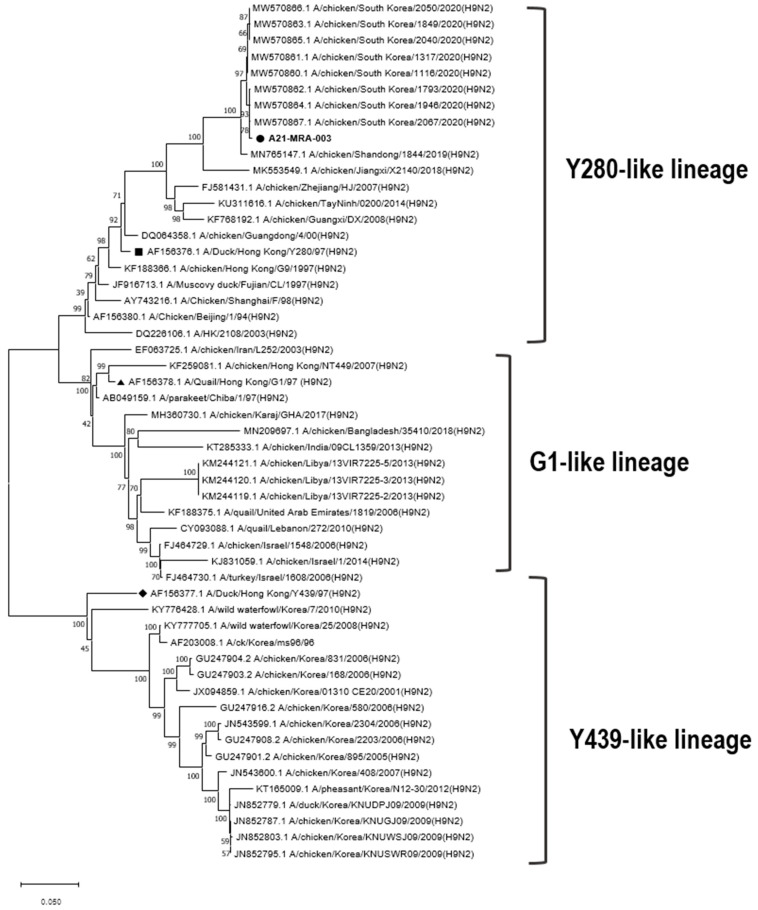
A phylogenetic analysis of H9N2/Y280 isolates using the haemagglutinin (*HA*) gene based on nucleotide sequences. Maximum likelihood phylogenetic analyses were conducted using MEGA-X software (www.megasoftware.net) with the Kimura 2-parameter model and 1000 bootstrap replicates.

**Figure 2 animals-14-00872-f002:**
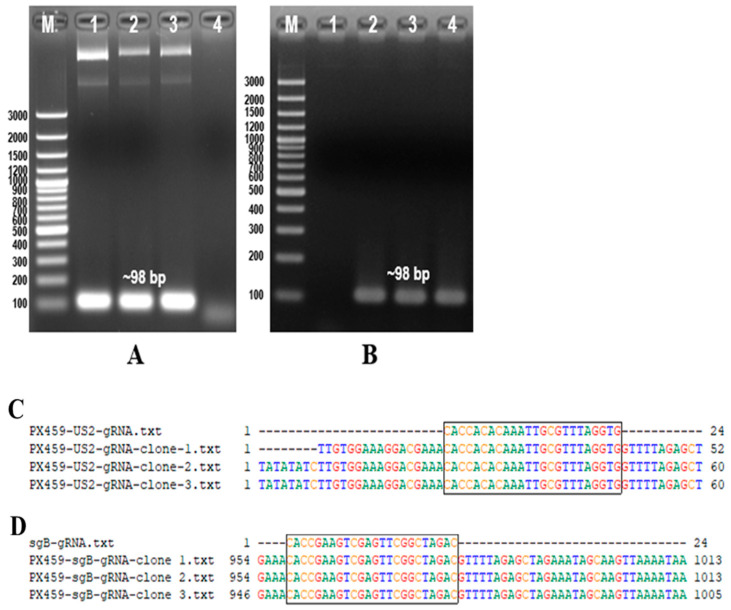
Identification of Cas9/gRNA and sequencing of Cas9/gRNA expression plasmids. (**A**) PCR identification of PX459-US2-gRNA using U6 forward and US2-gRNA-R primers. (**B**) PCR identification of PX459-sgB-gRNA using U6 forward and sgB-gRNA-R primers. (**C**) Sequencing of PX459-US2-gRNA with the U6 forward primer. (**D**) Sequencing of PX459-sgB-gRNA with the U6 forward primer.

**Figure 3 animals-14-00872-f003:**
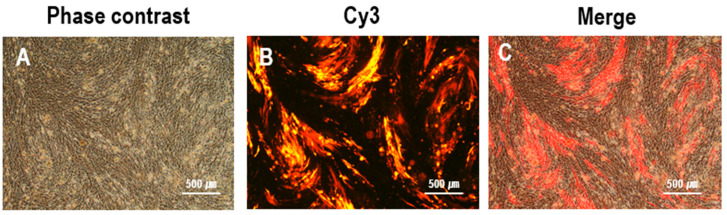
Observation of plaques expressing red fluorescence during rHVT/Y280 screening. rHVT/Y280-infected CEFs under phase contrast (**A**) and Cy3 excitation (**B**), and merge of both (**C**).

**Figure 4 animals-14-00872-f004:**
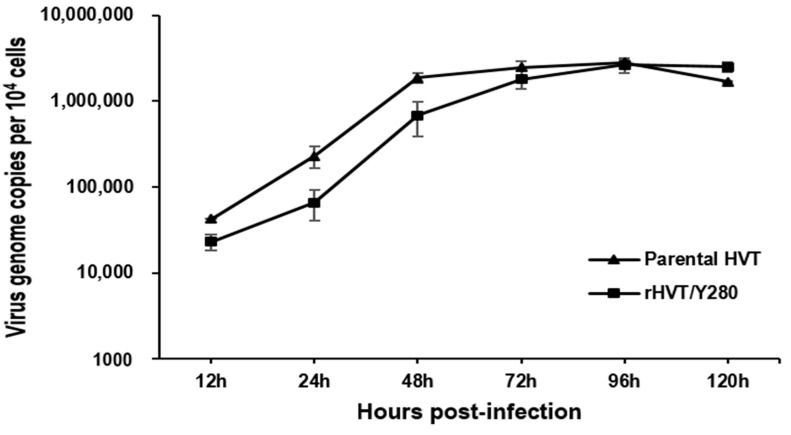
Growth kinetics of rHVT/Y280 and parental HVT. rHVT/Y280 and parental HVT viruses were inoculated into CEFs seeded in 24-well plates with 100 PFU/well. The cells were harvested and tested at 12, 24, 48, 72, 96, and 120 h post-infection.

**Figure 5 animals-14-00872-f005:**
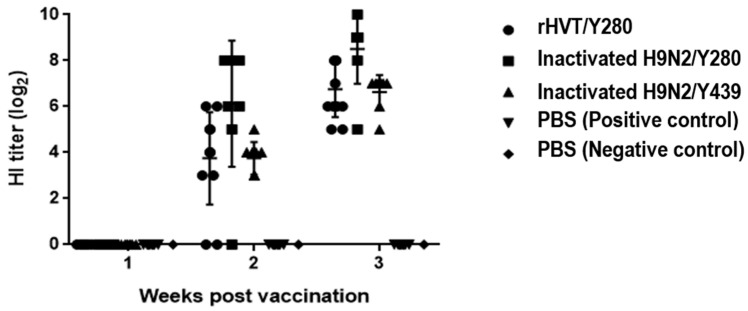
HI antibody analysis using sera collected weekly, as detected by HI assays.

**Table 1 animals-14-00872-t001:** Protective efficacy of the used vaccines against H9N2/Y280 challenge.

Group	Vaccination	Serology at 3 wpv (log_2_ HI titer) ^a^	Virus Shedding (log_10_EID_50_/0.1 mL) ^b^				Virus Replication in Tissues (log_10_EID_50_/0.1 mL) ^c^				
			3 dpi		5 dpi		5 dpi				
			OP	CL	OP	CL	Trachea	Lung	Bursa	Brain	Cecal Tonsil
A	rHVT/Y280	12/12 (6.8)	3/12 (0.5)	0/12 (−)	0/12 (−)	0/12 (−)	1/12 (0.1)	1/12 (0.1)	0/12 (−)	0/12 (−)	0/12 (−)
B	Inactivated H9N2/Y439	8/8 (8.5)	1/8 (0.3)	1/8 (0.2)	0/8 (−)	0/8 (−)	1/8 (0.1)	1/8 (0.2)	0/8 (−)	1/8 (0.2)	0/8 (−)
C	Inactivated H9N2/Y280	8/8 (6.9)	8/8 (4.0)	6/8 (1.4)	8/8 (1.8)	3/8 (0.6)	7/8 (2.0)	5/8 (1.4)	1/8 (0.2)	5/8 (1.2)	2/8 (0.5)
PC	PBS	0/10 (0)	10/10 (4.7)	8/10 (1.4)	10/10 (2.7)	9/10 (1.7)	10/10 (2.7)	7/10 (1.8)	5/10 (1.1)	6/10 (1.2)	6/10 (1.1)
NC	PBS	0/6 (0)	0/6 (−)	0/6 (−)	0/6 (−)	0/6 (−)	0/6 (−)	0/6 (−)	0/6 (−)	0/6 (−)	0/6 (−)

^a^ Number of serology-positive/total survivors in the group (mean HI titer). ^b^ Number of virus-positive/total in the group. ^c^ Tissue collected at 5 dpi, showing virus-positive/total in the group (virus titers from pooled samples).

## Data Availability

All the results of the study are presented within the manuscript and its Appendix A.

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
