# Peer review of "Protection of Chickens against H9N2 Avian Influenza Isolates with a Live Vector Vaccine Expressing Influenza Hemagglutinin Gene Derived from Y280 Avian Influenza Virus"

_animals, 2024, doi:10.3390/ani14060872_

Round 1

Reviewer 1 Report

Comments and Suggestions for Authors

The author used CRESPR/Cas9 technology to insert HA genes of AIV Y280 strain into HVT virus, and successfully constructed a recombinant HVT-HA vaccine. The effect was confirmed at the cellular and animal levels, demonstrating good immune efficacy. The research methods and results have certain reference significance for the development of AIV vaccines. Some details should be supplemented as following comments.

1. Page 3, The author need to label the 5 'and 3' of the primers, P3_For and P3_Rev.

2. Figure 3, The author should provide the magnification or scale bar of the image.

3. Figure 3, The author did not address the source and purpose of dye Cy3 in the Methods and Results, and reader cannot understand the meaning of stain Cy3 in Figure 3. Author should address the details of the application of Cy3 in Methods and the role of Cy3 in Results.

4. Figure S3, The order of genes could be reversed, but the labels should not be upside down. It is unreadable.

5. Most of citation is missing in Discussion section, citations should be added for all of the evaluative and conclusive sentences. 

6. The capitalization of unit symbols needs to be consistent throughout the text.

7. The author needs to add more details in Methods and Results.

Comments on the Quality of English Language

It is fine.

Author Response

Response to reviewer

Review 1

We would like to thank the reviewers for their helpful comments and suggestions, which were valuable in improving the quality of our manuscript. We have extensively and carefully revised the entire manuscript based on the review comments. The comments and the corresponding revisions made are listed below.

  1. Page 3, The author need to label the 5 'and 3' of the primers, P3_For and P3_Rev.

Answer: Thank you for the comment. As suggested, we have labeled the 5’ and 3’ of the primer sequences (lines 125–127).

  1. Figure 3, The author should provide the magnification or scale bar of the image.

Answer: Thank you for the comment. As suggested, we have revised Figure 3 (lines 299–301).

  1. Figure 3, The author did not address the source and purpose of dye Cy3 in the Methods and Results, and reader cannot understand the meaning of stain Cy3 in Figure 3. Author should address the details of the application of Cy3 in Methods and the role of Cy3 in Results.

Answer: Thank you for the comment. As suggested, we have addressed the source and purpose of dye Cy3 in Methods, Results, and Figure 3 (lines 170–180, 289).

  1. Figure S3, The order of genes could be reversed, but the labels should not be upside down. It is unreadable.

Answer: Thank you for the comment. As suggested, we have revised Figure S3.

  1. Most of citation is missing in Discussion section, citations should be added for all of the evaluative and conclusive sentences. 

Answer: As suggested, we have revised and added citations in the Discussion.

  1. The capitalization of unit symbols needs to be consistent throughout the text.

Answer: As suggested, we have revised this issue throughout the manuscript.

  1. The author needs to add more details in Methods and Results.

Answer: Thank you for the comment. As suggested, we have added more details in Methods and Results.

Reviewer 2 Report

Comments and Suggestions for Authors

The authors have submitted a high-quality manuscript for the novel generation of an effective live vaccine against low-path H9N2 avian influenza in chickens.  The results are strongly supported by data.  The discussion was articulate and meaningful.  The manuscript would benefit from some minor improvements to clarify some of the author's points.

Minor Findings:

1.  Explain the acronym HVT on its first use in the introduction.

2. In section 2.6, add a reference for the HI method or better explain the method in the section.

3.  For Figure 3, expand the figure legend to explain the microscopy or add this description in the materials and methods.

4.  In section 3.6 please list the clinical observations which were conducted as this information is not included in Table 1 or the materials and methods.

5.  Table 1 has a formatting issue.  The printed version shifts the header columns.

6.  In the discussion the authors describe the viral purification process and the utility of plaque selection vs. FACs.  Most of this discussion may be better suited for the Materials and Methods section.

7.  The authors assert the new live vaccine is better at inducing cellular and humoral immunity as compared to the current inactivated vaccine.  However, no data were presented on measures of cellular immunity.  The authors should clarify this assertion since no data were included or provide the data to support the assertion.

Author Response

Response to reviewer

Review 2

Comments and Suggestions for Authors

The authors have submitted a high-quality manuscript for the novel generation of an effective live vaccine against low-path H9N2 avian influenza in chickens.  The results are strongly supported by data.  The discussion was articulate and meaningful.  The manuscript would benefit from some minor improvements to clarify some of the author's points.

Minor Findings:

  1. Explain the acronym HVT on its first use in the introduction.

Answer: As suggested, we have added a sentence that explains the acronym HVT on its first use in the Introduction (line 81).

  1. In section 2.6, add a reference for the HI method or better explain the method in the section.

Answer: As suggested, we have revised the HI method in Section 2.6 Serology with reference that describes a standard protocol for this method (lines 235–243).

  1. For Figure 3, expand the figure legend to explain the microscopy or add this description in the materials and methods.

Answer: As indicated, we have added the description for Figure 3 (Light, Cy3, and Merged) in Section 2.3 Generation of recombinant rHVT/Y280 (lines 170–180, 289).

  1. In section 3.6 please list the clinical observations which were conducted as this information is not included in Table 1 or the materials and methods.

Answer: As suggested, we have added a new Section 2.9. Clinical signs, mortality, and gross lesions that describes the detailed clinical observation after challenges. Additionally, we have rearranged section numbers (lines 244–248).

  1. Table 1 has a formatting issue.  The printed version shifts the header columns.

Answer: Thank you for the comment. As pointed out, we have revised Table 1.

  1. In the discussion the authors describe the viral purification process and the utility of plaque selection vs. FACs.  Most of this discussion may be better suited for the Materials and Methods section.

Answer: Thank you for the comment. As per your suggestion, we have revised the Materials and Methods (Section 2.3. Generation of recombinant rHVT/Y280) (lines 170–180) and parts of the Discussion that describe plaque purification (lines 433–442).

  1. The authors assert the new live vaccine is better at inducing cellular and humoral immunity as compared to the current inactivated vaccine.  However, no data were presented on measures of cellular immunity.  The authors should clarify this assertion since no data were included or provide the data to support the assertion.

Answer: Thank you for your feedback. In the present study, we did not include the experiments on the cellular immune response following rHVT/Y280 vaccination. Thus, we have revised the part about cellular immunity throughout the manuscript. Considering that the HVT-vectored vaccine induces a strong cellular immune response, we have planned to test such rHVT/Y280-induced cellular immune response.

Reviewer 3 Report

Comments and Suggestions for Authors

Zhang et al. developed HVT-vectored vaccine for H9N2 low pathogenic avian influenza virus. The efficacy of the HVT-HA vaccine was demonstrated in specific-pathogen-free chickens. Vaccine was able to significantly reduced virus shedding, and inhibition of virus recovery in the cecal tonsil following challenge with a Korean Y280 H9N2 strain. These results support the potential of HVT vector vaccines expressing HA in protecting poultry against H9N2 viruses. However, the manuscript is poorly written and needs significant improvement to deliver its finding.

General comment:

l   English Editing: The manuscript requires comprehensive English editing for grammar, punctuation, and overall language clarity to meet the publication standards of a scientific journal.

l   Discrepancy in AIV Challenge Dose: There's a noted discrepancy between the challenge dose reported in the summary and the Materials & Methods section, which could lead to confusion and questions about the experimental accuracy.

l   Virus Shedding Detection: The decision to stop virus shedding detection at 5 days post-inoculation due to bird sacrifice overlooks the potential for longer shedding periods associated with LPAI, potentially affecting the assessment of vaccine efficacy against extended virus shedding.

Abstract:

l   The use of "genotype" instead of the correct term "lineage" for HA classification indicates a misunderstanding of virological terminology.

l   The emphasis on no fatalities among vaccinated chickens mislead readers that this Y280 H9N2 cause mortality.

l   Mentioning cellular immunity without presenting evidence weakens the scientific credibility of the reported findings.

Introduction:

l   The manuscript fails to expand abbreviations upon their first appearance, potentially confusing readers unfamiliar with specific terms.

Materials and Methods:

l   Omission of details such as the guide RNA design website, numerical units, the HI method, tissue sample processing, and specifics on vaccine preparation for certain groups detracts from the reproducibility and transparency of the study.

l   The discrepancy in the number of birds used in the experiment and their distribution into groups raises questions about the experimental design and data integrity.

Table 1:

l   Inconsistencies in the number of birds per group as presented in Table 1 further complicate the understanding of the study's experimental setup.

Comments on the Quality of English Language

The manuscript requires comprehensive English editing for grammar, punctuation, and overall language clarity to meet the publication standards of a scientific journal.

Author Response

Response to reviewer

Review 3

We would like to thank the reviewers for their helpful comments and suggestions, which were valuable in improving the quality of our manuscript. We have extensively and carefully revised the entire manuscript based on the review comments. The comments and the corresponding revisions made are listed below.

General comment:

l   English Editing: The manuscript requires comprehensive English editing for grammar, punctuation, and overall language clarity to meet the publication standards of a scientific journal.

Answer: Thank you for the comment. We have edited English to improve the overall language clarity throughout the manuscript. 

l   Discrepancy in AIV Challenge Dose: There's a noted discrepancy between the challenge dose reported in the summary and the Materials & Methods section, which could lead to confusion and questions about the experimental accuracy.

Answer: Thank you for pointing out this issue. Consequently, we have revised the challenge dose in the Simple summary and Conclusions (lines 20–21, 219, 466).

l   Virus Shedding Detection: The decision to stop virus shedding detection at 5 days post-inoculation due to bird sacrifice overlooks the potential for longer shedding periods associated with LPAI, potentially affecting the assessment of vaccine efficacy against extended virus shedding.

Answer: Thank you for your comment. Generally, H9N2/Y280 virus infection can induce long virus shedding until 7–10 dpi. Thus, we have planned to test virus shedding in long-term in a future study. In our study, we involved regulatory authorities for animal vaccine production in Korea for the calculation of protective indices based on cecal tonsil (CT). According to these regulatory authorities, the minimum efficacy requirement for inactivated H9N2 LPAI vaccines is >80% inhibition of the virus recovery rate from CTs in the vaccinated group compared to the recovery rate in unvaccinated control chickens at 5 days post-infection.

Song, J. M., Lee, Y. J., Jeong, O. M., Kang, H. M., Kim, H. R., Kwon, J. H., ... & Kim, Y. J. (2008). Generation and evaluation of reassortant influenza vaccines made by reverse genetics for H9N2 avian influenza in Korea. Veterinary microbiology130(3-4), 268-276.

Kim, D. Y., Kang, Y. M., Cho, H. K., Park, S. J., Lee, M. H., Lee, Y. J., & Kang, H. M. (2021). Development of a recombinant H9N2 influenza vaccine candidate against the Y280 lineage field virus and its protective efficacy. Vaccine39(42), 6201-6205.

Abstract:

l   The use of "genotype" instead of the correct term "lineage" for HA classification indicates a misunderstanding of virological terminology.

Answer: Thank you for your comment. We have revised the incorrect term “genotype” to “lineage” throughout the manuscript.

l   The emphasis on no fatalities among vaccinated chickens mislead readers that this Y280 H9N2 cause mortality.

Answer: We appreciate your feedback. We have revised the “no fatalities” to “no gross lesions” (line 33).

l   Mentioning cellular immunity without presenting evidence weakens the scientific credibility of the reported findings.

Answer: Thank you for indicating this issue. In the present study, we did not include experiments on the cellular immune response following rHVT/Y280 vaccination. Therefore, we have revised the part about cellular immunity throughout the manuscript.

Introduction:

l   The manuscript fails to expand abbreviations upon their first appearance, potentially confusing readers unfamiliar with specific terms.

Answer: Thank you for the comment. We have revised abbreviations throughout the manuscript.

Materials and Methods:

l   Omission of details such as the guide RNA design website, numerical units, the HI method, tissue sample processing, and specifics on vaccine preparation for certain groups detracts from the reproducibility and transparency of the study.

Answer: We appreciate your feedback. We have revised the methodology throughout the manuscript.

l   The discrepancy in the number of birds used in the experiment and their distribution into groups raises questions about the experimental design and data integrity.

Table 1:

 Answer: Thank you for the comment. Consequently, we have checked and revised the incorrect number of birds in Table 1 and throughout the manuscript.

l   Inconsistencies in the number of birds per group as presented in Table 1 further complicate the understanding of the study's experimental setup.

 Answer: Thank you for pointing out this issue. We have checked and revised the incorrect number of birds in Table 1 and throughout the manuscript.

Reviewer 4 Report

Comments and Suggestions for Authors

The authors provided a study on using a novel live vector vaccine to protect chickens against H9N2 Avian Influenza, showing promising results in inducing immunity and reducing virus shedding, suggesting its potential as an effective vaccine candidate.

1. Does the presence of RFP affect the growth of HVT-RFP-HA? It is crucial for the authors to present comparative data between the HVT-RFP-HA and the wild-type virus to clearly understand any variances in growth patterns.

2. What is the level of neutralizing antibodies generated by the HVT-RFP-HA?

3. TABLE 1 appears difficult to understand, especially the titles. Please make the necessary improvements.

Author Response

Response to reviewer

Review 4

We would like to thank the reviewers for their helpful comments and suggestions, which were valuable in improving the quality of our manuscript. We have extensively and carefully revised the entire manuscript based on the review comments. The comments and the corresponding revisions made are listed below.

  1. Does the presence of RFP affect the growth of HVT-RFP-HA?It is crucial for the authors to present comparative data between the HVT-RFP-HA and the wild-type virus to clearly understand any variances in growth patterns.

Answer: Thank you for the comment. We have already tested the growth property of recombinant virus compared with this parental virus (wild-type virus), but we did not include the respective data in the manuscript. We revealed that the recombinant virus that inserted RFP-H9/Y280 did not show a statistically significant difference compared to the parental virus at every time point. Thus, the presence of RFP-H9/Y280 (including the RFP expression cassette) did not affect viral growth. Following your suggestion, we have added data for the growth kinetics of rHVT/Y280 in the manuscript (lines 182–195, 310–319, Figure 4).

  1. What is the level of neutralizing antibodies generated by the HVT-RFP-HA?

Answer: Haemagglutinin inhibition (HI) titers described in the manuscript are the level of neutralizing antibodies generated by HVT-RFP-HA (rHVT/Y280).

HI titers measured by the HI assay represent neutralizing antibody titers against avian influenza virus (AIV). Haemagglutinin (HA) glycoprotein is the major viral surface protein with chicken red blood cell–binding capacity. Thus, HI assay applied this principle to measure neutralizing antibody titers (HI titers). In our study, we inserted the full H9N2/Y280-HA-encoding gene with an expression cassette in HVT. As a result, our recombinant rHVT/Y280 could express the H9N2/Y280-HA protein, inducing H9N2/Y280-HA protein–neutralizing antibodies in rHVT/Y280-vaccinated chickens.

  1. TABLE 1 appears difficult to understand, especially the titles. Please make the necessary improvements.

Answer: Thank you for indicating this issue. Consequently, we have checked and revised the incorrect number of birds in Table 1 and throughout the manuscript.

Round 2

Reviewer 3 Report

Comments and Suggestions for Authors

The authors addressed all the issues raised by reviewers.

Keep up the good work. Looking forward to seeing real world application of the advanced vaccine technology for poultry health.